# A 30-Year Experience in Fragile X Syndrome Molecular Diagnosis from a Laboratory in Thailand

**DOI:** 10.3390/ijms26157418

**Published:** 2025-08-01

**Authors:** Areerat Hnoonual, Oradawan Plong-On, Duangkamol Tangviriyapaiboon, Chariyawan Charalsawadi, Pornprot Limprasert

**Affiliations:** 1Department of Pathology, Faculty of Medicine, Prince of Songkla University, Songkhla 90110, Thailand; areerat.h@psu.ac.th (A.H.); oradawan@hotmail.com (O.P.-O.); cchariya@yahoo.com (C.C.); 2Genomic Medicine Center, Faculty of Medicine, Prince of Songkla University, Songkhla 90110, Thailand; 3Rajanagarindra Institute of Child Development, Chiang Mai 50180, Thailand; dtangviriyapaiboon@gmail.com

**Keywords:** fragile X syndrome, *FMR1*, full mutation, molecular diagnosis

## Abstract

Fragile X syndrome (FXS) is the most common form of X-linked intellectual disability (ID). This study aimed to share 30 years of experience in diagnosing FXS and determine its frequency in Thailand. We retrospectively reviewed 1480 unrelated patients (1390 males and 90 females) with ID, developmental delay, or autism spectrum disorder, or individuals referred for FXS DNA testing at Songklanagarind Hospital, Thailand, over a 30-year period. The samples were analyzed using cytogenetic methods, PCR-based techniques, and/or Southern blot analysis. Full mutations (>200 CGG repeats) were identified in 100 males (7.2%) and three females (3.3%). An intermediate allele was detected in one male, while no premutation was found in the index cases. Two males were suspected to have *FMR1* gene deletions. Twelve families underwent prenatal testing during this study. Most families undergoing prenatal FXS diagnosis involved mothers who were premutation carriers and had given birth to children affected by FXS. This study represents the largest series of molecular genetic FXS testing cases reported in Thailand. The frequency of FXS identified in different cohorts of Thai patients across various periods was approximately 7%. This study enhances public awareness of at-risk populations and highlights the importance of prenatal testing and genetic counseling for vulnerable families.

## 1. Introduction

Fragile X syndrome (FXS; MIM 300624) is the most common form of X-linked intellectual disability (ID) and the most common monogenic cause of autism spectrum disorder (ASD). It is characterized by neurobehavioral features such as developmental delay (DD), learning disabilities (LDs), and attention deficit hyperactivity disorder (ADHD), as well as physical hallmarks of FXS, including an elongated face, large or prominent ears, and macroorchidism [1]. Male patients are more frequently affected than female patients and have greater disease severity. Non-random X-chromosome inactivation (XCI) may explain the milder phenotype observed in females with FXS, with phenotypic variability depending on the degree of non-random XCI [2].

FXS is caused by the expansion of CGG repeats in the 5′-untranslated region of the fragile X messenger ribonucleoprotein 1 (*FMR1*) gene. Although FXS is usually caused by the expansion of the CGG repeat of *FMR1*, rare mutations, including point mutations or deletions, leading to FXS, have been reported [3]. FXS is mainly caused by a repeat expansion mutation in the *FMR1* gene, which is usually inherited from the mother, although rare cases of de novo mutations causing FXS have been reported [3,4]. The *FMR1* gene encodes the FMR1 protein (FMRP), which is essential for normal brain development. Individuals with over 200 CGG repeats were considered to have full mutations, whereas those with 5–44 repeats were considered within the normal range. Females with a premutation allele, characterized by 55–200 CGG repeats, or intermediate alleles with 45–54 CGG repeats, do not have FXS. However, they are at an increased risk of expansion to full mutations and premutations in subsequent generations. According to the American College of Medical Genetics and Genomics (ACMG) standards and guidelines for fragile X testing, prenatal diagnosis should be considered in future pregnancies for individuals carrying intermediate or premutation alleles [2,5]. The risk of expansion from female premutation carriers to full mutation in subsequent generations depends on the size of the CGG repeats, with larger repeats carrying a greater risk of expansion. The presence of AGG interruptions every 9- or 10-CGG repeat block increases allele stability during transmission [6,7,8]. Individuals with premutation alleles are usually unaffected by FXS. However, female premutation carriers are at an increased risk of developing fragile X-associated primary ovarian insufficiency (FXPOI). Moreover, both males and females with premutations are at risk of fragile X tremor/ataxia syndrome (FXTAS) and fragile X-associated neuropsychiatric disorders (FXANDs) [9]. Females are more frequent carriers of the premutation allele than males are. The reported premutation carrier frequencies range from 1 in 250 to 1 in 850 in males and from 1 in 110 to 1 in 300 in females, depending on the population [10,11]. The frequency of fragile X premutations in the Thai female population was recently reported to be 1 in 540 (0.19%) [12,13].

The estimated frequency of the fragile X full mutation is approximately one in 7000 males and one in 11,000 females across various ethnic groups [10]. Large-scale studies conducted in various populations have reported a wide range of FXS frequencies in patients with ID, DD, and/or ASD [14,15,16,17,18,19,20,21,22,23], possibly due to variations in genetic backgrounds and other factors affecting sample inclusion. The burden of FXS in Thailand has received little attention in the scientific literature. Research on FXS in Thailand remains sparse because of limited awareness, diagnostic services, and trained personnel. The frequencies of FXS in Thai patients with ID from Southern and Central regions have been reported to be 6.8% and 7.5%, respectively [24]. A previous study reported that neither premutations nor full mutations were identified in Thai patients with ASD of unknown etiology [25]. However, all patients with FXS were likely not completely accounted for in Thailand, as populations from other regions were excluded. In addition, the genetic diversity of the Thai population may affect the frequency and presentation of FXS in this region. As the diagnostic center for FXS in Thailand, receiving samples from all regions of the country, this study shares 30 years of experience in diagnosing FXS, with a particular emphasis on both postnatal and prenatal FXS diagnosis, and aimed to determine the frequency of FXS in high-risk groups in Thailand. Furthermore, we reviewed the literature on the frequency of FXS in patients with ID, DD, and ASD in various populations worldwide.

## 2. Results

### 2.1. Frequency of FXS

We retrospectively reviewed FXS in 1480 unrelated Thai patients, comprising 394 male patients with ID across three pediatric cohorts, and 1086 patients referred for routine FXS testing. Among male patients with ID across pediatric cohorts, 132 were from Southern Thailand, 161 from Central Thailand, and 101 from Northern Thailand. Fragile X full mutations were identified in 29 patients, including nine from Southern Thailand (9/132, 6.8%), twelve from Central Thailand (12/161, 7.5%), and eight from Northern Thailand (8/101, 7.9%). The frequency of FXS in Thai males with ID was 7.4% (29/394) in this study.

Since 2000, fragile X testing has been offered as a routine service at Songklanagarind Hospital, which is the main laboratory for FXS testing in Thailand. Between 2000 and 2021, 1086 patients, including 996 males and 90 females, were referred to our center for FXS testing. The ages of the patients ranged from 0.2 to 15 years. This cohort of patients originated from all regions of Thailand, with the majority originating from the Southern and Central regions. We identified 1008 patients (921 males and 87 females) with CGG repeat numbers within the normal range. One male patient had intermediate CGG repeats (47 repeats), and no premutation alleles were identified among the index cases in this cohort. Full mutations were identified in 71 male (71/996; 7.1%) and three female (3/90; 3.3%) patients. Among the full mutation males, mosaicism was detected in six individuals, including five males with both premutation and full mutation alleles and one male with mosaic normal, premutation, and full mutation alleles. Furthermore, two male patients with suspected *FMR1* deletions were identified among the FXS-positive index cases in this cohort. Moreover, three patients who underwent FXS testing had sex chromosome aneuploidies (two cases of 47,XXY and one case of 46,XY in a female), which were confirmed by standard karyotyping.

The overall frequency of full mutations across all cohorts in this study was estimated to be 7% (103/1480), comprising 7.2% (100/1390) of males and 3.3% (3/90) of females. The results of molecular diagnoses of FXS conducted between 1991 and 2021 are summarized in Table 1.

Normal alleles were identified in the distribution of CGG repeats, with sizes ranging from 5 to 44. In this study, the most frequent allele among individuals with a normal repeat range of *FMR1* was 29 CGG repeats (53.6%), followed by 30 CGG repeats (22.7%), and a minor allele with 36 CGG repeats (8.6%). The distribution of normal *FMR1* alleles from male patients with a single normal allele is presented in Appendix A.

### 2.2. Prenatal FXS Diagnosis

Prenatal testing for FXS was conducted in 12 families with complete results over a 30-year period (Table 2). In our laboratory, prenatal testing was performed on DNA extracted from cultured amniotic fluid cells. PCR-based methods and/or Southern blot analysis were used to determine the CGG repeat size and the methylation status of the fetus. The most undergoing prenatal FXS diagnosis involved mothers who were premutation carriers and had given birth to children affected by FXS. In an exceptional family (F4), the mother had *FMR1* alleles within the normal range, whereas the father carried a premutation allele inherited from his mother, who was also a carrier of the premutation. Although paternal premutation alleles in the *FMR1* gene are generally less likely to expand to full mutation compared to maternal premutation alleles, this family requested prenatal FXS testing. The second fetus in this family showed a slight change in CGG repeat numbers, although the repeats remained within the premutation range. For prenatal testing, two unique cases were identified: in the F20 family, the mother carried the full mutation allele, and in the F12-2 family, the mother exhibited a mosaic pattern with both premutation and full mutation alleles. In the F12-2 family, the mother had three children with different *FMR1* genetic profiles. The first child had a mosaic premutation and a de novo 313 bp deletion of the *FMR1* gene. The second child was a male with a full mutation, and prenatal testing for the third child revealed a male with a normal CGG repeat allele. This family has been reported in our study [3].

### 2.3. Full Mutation Expansion in Maternal Transmission

In patients with fragile X full mutation, mothers must typically undergo *FMR1* status testing, which can be helpful for management and family planning. Among the mothers of children with full FXS mutations, 30 individuals with available DNA were tested for *FMR1* gene status. In this cohort, 28 of 30 mothers of children diagnosed with full mutation were confirmed to be premutation carriers. However, one mother was confirmed to be a full mutation carrier with random X-inactivation (29/>200 CGG repeats). Her son was a full mutation male, and her daughter was a full mutation female with suspected skewed X-inactivation, as only the methylated allele was detected using Southern blot analysis. Another mother was identified as having mosaic premutation and full mutation (37/~140/>200 CGG repeats), which has been previously reported [3]. None of the mothers carried alleles containing <70 CGG repeats. Among the 34 children with full mutations whose mothers were tested for AGG interruptions, TP-PCR analysis revealed 25 full mutation transmissions from maternal alleles with no AGG interruptions and nine transmissions from premutation alleles with one AGG interruption. None of the mothers with premutations had more than one AGG interruption. In this study, the smallest maternal allele expanding to full mutation was 70 CGG repeats, with no AGG interruptions. In this cohort, maternal alleles with either no AGG interruptions or a single AGG interruption were associated with the expansion of CGG repeats from the premutation to full mutation range in their offspring (Appendix A).

## 3. Discussion

We retrospectively reviewed FXS in 1480 unrelated Thai patients with ID, DD, or ASD across four cohorts, utilizing a large and representative sample covering all regions of Thailand. Our study revealed an overall full mutation frequency of 7%, including 7.2% in males and 3.3% in females, providing valuable insights into the frequency of FXS in this population. The frequency of FXS varies among different populations. Large cohort studies on the frequency of FXS have predominantly been conducted in European and American populations, whereas studies on Asian populations are mostly on a smaller scale [26]. Asia is known for its genetic diversity, with populations across various regions of the continent showing differences in the frequency of FXS. Based on our literature review, a notably high frequency of FXS among individuals with ID has been reported in populations from the Middle East, with the highest frequency (11%) of full mutation observed in Kuwaiti patients [27], potentially due to the founder effect and consanguinity. In contrast, East Asian populations have reported a low frequency of FXS, ranging from 0% to 2.8% [14,28,29,30,31,32,33,34,35]. Very few reports on the frequency of FXS have been published regarding Southeast Asian populations. Previous research conducted in Indonesia reported a frequency of FXS of 4% among patients with ID [36] and 1.5% among patients with DD [37]. In Malaysia, among patients with LD from mixed ethnicities, the full FXS mutation was reported in 3.5% [23]. The variation in the reported frequency of fragile X mutations across different Asian studies could be attributed to either selection criteria based on the clinical characteristics of FXS or the influence of a founder effect [38]. However, FXS frequency data from Southeast Asia is sparse and often limited to small-scale studies, highlighting the need for more comprehensive research in this area.

### 3.1. Frequency of FXS in Males

In this study, the frequency of FXS in males was found to be 7.2% (100/1390). Overall, this study revealed a frequency higher than previously reported in male patients with ID, DD, and/or ASD in other Southeast Asian populations (1.1–3.5%) [23,36,37,39], East Asia (0.4–2.9%) [28,29,30,32,33,34,35,40], and in studies from the USA with mixed populations (0.3–1.4%) [20,41,42,43]. Studies in Southeast Asia include a cohort study in Indonesia that found full mutation in 1.1% (1/92) of males with ID [36], comparable to another study reporting FXS in 1.9% (4/206) of males patients with DD [37]. In Malaysia, full FXS mutation was reported in 3.5% (73/2057) of males with LD [23]. A frequency rate of 2.4% (6/255) was observed among male patients with LD in Singapore [39]. The higher frequency observed in our study may be due to the selection bias of patients referred for routine FXS diagnosis based on the most prominent criteria for FXS, which increases the likelihood of identifying individuals with FXS.

However, these findings on the frequency of fragile X full mutations in Thai male patients are comparable to those reported in South Asia (2–9.7%) [19,38,44,45,46], the Middle East (7.2–11%) [16,27], and Africa (4.8–6.1%) [18,47,48]. A considerable variation in FXS frequency was noted among male patients within European high-risk populations, ranging from 0.5% to 6.5% [15,17,49,50,51,52,53,54,55,56,57,58,59,60] (Figure 1 and Appendix A).

### 3.2. Frequency of FXS in Females

The frequency of the fragile X full mutation in Thai females was 3.3% (3/90), which is consistent with several previous studies [18,23,28,29,30,45,46,49,57]. In contrast, higher frequency rates (7.1%, 6/84) were observed among Indonesian patients with ID, ASD, and/or a family history of FXS [36]; however, this may be due to sample bias, as only a small population of female patients was included in this study. Among females with ID or DD, previous studies have reported varying frequencies, ranging from 0% to 7.1%, across different populations.

The reported frequency of FXS in female patients is consistent with males being more affected than females by FXS. However, fewer female than male patients were included in each study, which may not accurately reflect the actual frequency of FXS in females within the population. Although FXS can be explicitly diagnosed, females often exhibit milder features, making early diagnosis more challenging, particularly in regions with limited access to genetic testing, geneticists, and specialized pediatricians. Despite the availability of FXS testing in Thailand for over 30 years, only a few female patients suspected of having FXS have been referred for diagnostic testing. Comparing the frequency of FXS among populations is difficult because the variation in FXS frequency reported in previous studies on high-risk populations is likely due to differences in the populations studied, sample sizes, and diagnostic methods. For instance, PCR followed by gel electrophoresis may not accurately determine the CGG repeat size and may not distinguish between large premutations and full mutations, demonstrating the potential limitations of certain diagnostic methods.

### 3.3. Distribution of CGG Repeats

The distribution of CGG repeat sizes varies among different populations. In this study, the most common normal CGG repeat allele in the Thai population was 29, followed by 30 and 36. This finding is consistent with studies in other Asian populations, where 29 CGG repeats have been observed more frequently than 30 CGG repeats [61,62,63]. However, some studies on Asian populations have identified the most common alleles, including 28 CGG repeats in Indian populations [64] and 27 CGG repeats in the Japanese population [32]. In contrast, the most common allele among African American populations was 30 CGG repeats, followed by 29 and 31 repeats [41]. Similarly, the most common allele in European populations is 30 CGG repeats [41,65,66,67]. Directly comparing the frequency distribution of CGG repeats across studies is challenging because of differences in race and ethnicity among populations and variations in the methods used to determine repeat sizes [2,68]. For example, PCR-based methods may affect repeat determination accuracy, whereas capillary electrophoresis provides more precise repeat sizing than gel electrophoresis [20]. Since 2006, nonradioactive PCR has been converted to fluorescent PCR, followed by capillary electrophoresis using an automated DNA sequencer in our laboratory to improve the accuracy of CGG repeat size determination.

### 3.4. Prenatal FXS Diagnosis

In this study, despite having 30 years of experience in FXS testing, participation was limited, with only 12 families undergoing prenatal testing during this period in our laboratory. The main reason for prenatal FXS diagnosis is a positive family history. This low number starkly contrasts with the numerous patients with FXS identified, indicating significant gaps in family investigations. This may be due to limited awareness, genetic counseling, and diagnostic services. A major reason for this could be the limited availability of genetic counseling services, which play a crucial role in informing and encouraging families to consider further genetic testing. Although most women were aware of the risk of Down syndrome, few understood the implications of FXS. Previous studies have revealed that 64% of pregnant women with a positive family history of FXS are unaware of their status and have at least one child affected by FXS before undergoing prenatal diagnosis. Consequently, prenatal and maternal tests are performed simultaneously [69]. Most medical professionals do not have firsthand experience with FXS families [70]; therefore, they may not be fully aware of the pathogenesis and disease transmission of FXS. Studies on attitudes toward prenatal screening and testing for FXS in pregnant women indicate limited pre-test knowledge about FXS, with only 33% having heard of it before enrollment in this study. Even after counseling, knowledge remains limited, with only 30% of participants accurately understanding the 50% risk of each female child inheriting their condition [71]. This highlights the need for improved education and counseling for families with FXS, particularly during prenatal testing. Ensuring that families have accurate and comprehensive information about the condition and its transmission can aid in better decision-making and understanding of potential risks. Thailand has only a few genetic counselors, most of whom work in tertiary hospitals. Consequently, many medical professionals may not be familiar with FXS, and the full genetic counseling training program is not currently officially established in the country. However, an increasing number of genetic counselors has been the focus of government policy since 2022, driven by the Genomic Thailand Project.

Furthermore, the limited number of laboratories in Thailand that can conduct FXS diagnostic testing presents a significant barrier. Only two laboratories offer postnatal FXS testing services, and our laboratory is the only one providing prenatal FXS testing at this time. This restriction limits access to essential diagnostic facilities, particularly for populations residing in remote or underserved areas. Although commercial kits for *FMR1* CGG repeats are available, their widespread application is restricted by the high cost of equipment and kits, the complexity of the results, and the need for skilled inspectors. Based on our experience, performing prenatal testing for FXS is difficult due to limitations in the quantity and quality of DNA extracted from amniotic fluid. Moreover, the interpretation and genetic counseling of the results for prenatal FXS testing can be particularly challenging, especially in cases of fetuses with mosaicism or female fetuses with a full mutation. These challenges underscore the need to expand genetic counseling efforts and enhance laboratory capacities to better serve affected families and prevent fragile X expansion transmission.

### 3.5. Full Mutation Expansion in Maternal Transmission and AGG Interruptions

To assess the impact on repeat instability and full mutation expansion, the status of the *FMR1* gene and the number of AGG interruptions in the mothers of patients with full mutations were determined. As a limitation of our study, not all mothers of cases with full mutations were tested for their *FMR1* status due to limited access to genetic counseling and financial constraints faced by each family. Among the available samples in this cohort, none of the tested mothers had an intermediate or low range of premutations (<70 CGG repeats). All tested mothers carried either premutation or full mutation alleles. This is not surprising because mothers who underwent AGG testing typically discovered their premutation status after having a child diagnosed with FXS. All full mutation expansions occurred in maternal alleles with either no AGG interruption or only one AGG interruption. Alleles with no AGGs exhibit the greatest degree of instability during transmission, and maternal premutation alleles with no AGGs have the greatest risk of expanding to a full mutation [8]. Although higher CGG repeat numbers indicate a higher risk of expansion, AGG interruption is also an important factor for predicting CGG repeat instability [8,72,73]. Determining the number of AGG interruptions in female premutation carriers with 55–90 CGG repeats can modify the risk of expansion to a full mutation, which can inform reproductive decision-making. A large difference in reproductive risk among premutation carriers with 75–80 CGG repeats, depending on the AGG interruption, has been reported [7,74]. However, determining AGG interruptions is not recommended for women with >90 CGG repeats because the risk of expansion to a full mutation in the offspring is almost 100% [8,20,75]. In our FXS laboratory in Thailand, carrier testing was performed according to the current guidelines for women with a history of ovarian insufficiency, family history of FXS or related disorders, unexplained neuropsychiatric symptoms, or premature ovarian insufficiency [76]. AGG interruptions were tested in individuals with intermediate and premutation alleles to determine the risk of expansion.

### 3.6. Molecular Diagnosis of FXS

In our cohort of patients referred for FXS DNA testing with a normal CGG repeat range, three were identified with sex chromosome abnormalities, which were confirmed through standard karyotyping. This finding supports the recommendation to consider chromosome abnormality testing along with FXS testing in patients with ID and DD. Currently, chromosomal microarray, whole exome sequencing, and whole genome sequencing can increase the diagnostic yield in these groups of patients [77,78]. Molecular diagnostic tests for FXS include PCR amplification of CGG repeat regions, MS-PCR, and Southern blot analysis. Currently, no single method can be universally applied for the laboratory diagnosis of FXS [2]. A comprehensive approach combining multiple molecular techniques is essential for an accurate and reliable diagnosis. PCR estimates CGG repeat size, especially in the normal and premutation ranges [2]. However, PCR can be problematic when amplifying large premutation alleles and may not amplify full mutation alleles. MS-PCR is a sensitive and rapid method for detecting the methylation status of the *FMR1* gene, particularly when only a little DNA is available. Although the methylation status of a gene can be rapidly determined using MS-PCR, Southern blot analysis can distinguish females with premutations or full mutations from normal females, which cannot be achieved using MS-PCR alone.

Southern blot analysis remains the gold standard for the molecular diagnosis of FXS. Since 1997, the fragile X expansion mutation has been confirmed in our laboratory using the standard Southern blotting method. In our laboratory, two cases of suspected *FMR1* deletions were confirmed using Southern blot analysis, whereas other methods showed inconclusive results. This suggests that PCR-based methods cannot replace Southern blotting in complex cases. Recently, TP-PCR and msTP-PCR methods have been developed to detect both full mutation allele and low-level mosaicism [79,80,81]. In cases of insufficient DNA, especially in prenatal cases where Southern blotting cannot be performed, we used other methods, including TP-PCR, MS-PCR, and msTP-PCR, to confirm both the CGG repeat and methylation status. In addition, the number and position of AGG interruptions detected using TP-PCR or msTP-PCR can be used to assess the risk of expansion upon transmission, especially in female carriers of intermediate and premutation alleles. Newer technologies, such as long-read DNA sequencing, are an option for FXS testing, particularly for patients with suspected deletions or point mutations of the *FMR1* gene that may be missed by conventional methods used for routine diagnosis [82]. However, future studies are needed to optimize the method, and cost-effectiveness should be considered for routine use in clinical practice. Some suspected cases of FXS may require multiple testing methods because of inconclusive or ambiguous results.

## 4. Materials and Methods

### 4.1. Cohort in This Study

A total of 1480 patients referred for FXS DNA testing, including 1390 males and 90 females, were recruited from four cohorts between 1991 and 2021. A retrospective review was conducted on 293 males with ID of unknown cause from previous research studies conducted between 1991 and 1999 [24] and 101 males with ID from a psychiatric clinic in Northern Thailand during 2007–2008. In addition, 1086 unrelated patients (996 males and 90 females) referred for FXS DNA testing at the Songklanagarind Hospital Laboratory, which serves as the central laboratory for FXS testing in Thailand, from 2000 to 2021 were included in this study. The inclusion criteria for patients referred for FXS DNA testing were Thai children aged 15 years or younger with ID, DD, and/or ASD.

This study’s protocol was approved by the Institutional Ethics Committee of the Faculty of Medicine, Prince of Songkla University (REC. 64-435-5-2).

### 4.2. FXS DNA Analysis

DNA was extracted from peripheral blood or amniotic fluid samples using the standard phenol/chloroform method or the FlexiGene DNA kit (QIAGEN, Hilden, Germany), following the manufacturer’s instructions. FXS testing was performed using cytogenetic analysis between 1991 and 1996 and a molecular diagnostic approach from 1996 to the present. The techniques used to diagnose FXS in our laboratory were based on an appropriate period (Figure 2). Cytogenetic analysis of the fragile X site at Xq27.3 was performed at our laboratory, as previously described [83]. All cases diagnosed using cytogenetic methods were confirmed using DNA analysis.

Various PCR-based methods have been used in our laboratory, including nonradioactive PCR, fluorescent PCR, methylation-specific PCR (MS-PCR), triplet-repeat-primed PCR (TP-PCR), and methylation-specific triplet-primed PCR (msTP-PCR). Depending on the patient’s sex, we used more than one method for FXS testing, following the ACMG standards and guidelines [2]. Briefly, samples from male patients were tested using PCR for CGG repeat sizing (i.e., PCR across the CGG repeat using flanking primers) either via conventional nonradioactive PCR or fluorescent PCR. Samples that failed to amplify were subsequently confirmed using MS-PCR. In female patients, besides PCR for CGG repeat sizing, TP-PCR was used to distinguish females with premutation and full mutation alleles from those with normal homozygous alleles. msTP-PCR was used to confirm CGG repeat expansions and the methylation status of the *FMR1* gene in cases with unclear results or insufficient DNA for Southern blot analysis. Southern blot analysis, the gold standard for FXS diagnosis, was used to confirm positive and ambiguous PCR results.

#### 4.2.1. Nonradioactive PCR

Nonradioactive PCR was used to estimate the CGG repeat length in the promoter region of the *FMR1* gene. The PCR protocol and primer sequences have been described previously.

#### 4.2.2. MS-PCR and msTP-PCR

MS-PCR and msTP-PCR were performed to determine the methylation status of the *FMR1* promoter region. Genomic DNA was treated with sodium bisulfite, followed by MS-PCR. For male samples, MS-PCR was performed using primer sets designed by Weinhausel et al. [84], following a previously described protocol [13,85]. For female DNA samples, msTP-PCR was used to confirm the repeat expansion and methylation status of the *FMR1* gene [3,86]

#### 4.2.3. Fluorescent PCR

Fluorescent PCR was performed on all participants to determine the size of the CGG repeats. This method can detect *FMR1* normal and low-range premutation alleles. In our laboratory, nonradioactive PCR has been replaced by fluorescent PCR since 2006. Forward (FRAXA-PSU: 5′-6FAM-CAGCGTTGATCACGTGACGTGGTTTCAGTG-3′) [25] and reverse primers (Primer 3: 5′-GTGGGCTGCGGGCGCTCGAGG-3′) [87] were used for PCR amplification, followed by capillary electrophoresis.

PCR amplification was performed in a final reaction volume of 10 μL, containing 25 ng of genomic DNA, 1× ImmoBuffer (Bioline, Taunton, MA, USA), 0.2 mM of dNTP with 100% of 7-deaza-dGTP, 1.5 mM of MgCl_2_, 0.2 μM of each primer, 2.2 M of betaine, and 0.5 units of Taq DNA polymerase (Immolase, Bioline). The PCR conditions included initial denaturation at 95 °C for 5 min, followed by 10 cycles at 95 °C for 35 s, 64 °C for 35 s, and 72 °C for 2 min. This was followed by 25 cycles of 95 °C for 35 s, 64 °C for 35 s, and 72 °C for 2 min, with a 10 s increment for each cycle, and a final extension at 72 °C for 10 min. The PCR products were analyzed using capillary electrophoresis using a Genetic Analyzer 3130 or 3500 (Applied Biosystems, Waltham, MA, USA). Briefly, 1 μL of the PCR product was mixed with 10.6 μL of Hi-Di Formamide and 0.4 μL of GeneScan-600 LIZ size standards. The samples were denatured at 95 °C for 2 min and cooled to 4 °C for 5 min before loading into the capillary electrophoresis system. The data were analyzed using GeneMapper software (version 6.0).

#### 4.2.4. TP-PCR

TP-PCR of the *FMR1* gene was used to detect the presence of premutations and/or full mutations in samples in which fluorescent PCR results suggested repeat expansions. TP-PCR was performed to distinguish between females with normal homozygous alleles and those with expanded alleles. TP-PCR was also used to determine AGG interruption patterns within CGG repeats in individuals with repeat expansions.

The PCR reactions were conducted in 15 µL mixtures containing 100 ng of genomic DNA, 1X ImmoBuffer (Bioline), 2.5 mM MgCl_2_, 400 µM dNTPs (with dGTP replaced by 7-deaza dGTP), 2.2 M betaine, 0.3 µM of each c primer (5’-6FAM-GCTCAGCTCCGTTTCGGTTTCACTTCCGGT-3’) and CCG-chimeric primer (5’-AGCGTCTACTGTCTCGGCACTTGC(CCG)_4_-3’) [88], and 1.5 units of HotStarTaq DNA Polymerase (Qiagen, Germany). The PCR reactions were run on the GeneAmp PCR System 9700 (Applied Biosystems) using mode 9700 (maximum ramp rate of 5 °C/s). The PCR cycling profile was as follows: denaturation at 95 °C for 15 min, 45 cycles of 95 °C for 45 s, 59 °C for 35 s, and 72 °C for 4 min, followed by a final extension at 68 °C for 10 min. One microliter of the PCR product was then mixed with 10.7 µL of Hi-Di formamide and 0.3 µL of LIZ600, denatured at 95 °C for 2 min, and cooled at 4 °C for 5 min. The PCR mixture was subsequently loaded onto an ABI3500 Genetic Analyzer and analyzed using GeneMapper software.

#### 4.2.5. Southern Blot Analysis

Southern blot analysis was used to confirm cases with suspected premutations and/or full mutations. Southern blotting was performed as previously described [24]. Briefly, 8–10 µg of genomic DNA was digested using EcoR I and Eag I. Following digestion, the samples were hybridized with a probe specific to the *FMR1* gene (StB12.3), which was labeled with alkaline phosphatase for subsequent detection [24].

In this study, according to the guidelines of the ACMG [2], *FMR1* alleles were classified as normal (≤44 CGG repeats), gray zone or intermediate (45–54 CGG repeats), premutation (55–200 CGG repeats), or full mutation (>200 CGG repeats).

### 4.3. Data Collection and Search Strategy for Literature Review

We conducted a comprehensive narrative review to investigate the frequency of FXS in high-risk groups worldwide using molecular testing. Relevant articles were searched using two main medical databases, PubMed and Web of Science, with the primary keyword “fragile X syndrome,” combined with other keywords: “*FMR1*,” “prevalence,” “frequency,” “intellectual disability,” “developmental delay,” “autism spectrum disorder,” “high-risk,” and “molecular testing.” Only articles that were accessible and published in English were selected for the review. No time limit was set for the published literature. Additional studies were identified through manual screening of reference lists.

## 5. Conclusions

This study represents the largest cohort of patients referred for FXS testing in Thailand over a 30-year period. The frequency of FXS in the high-risk population of Thailand is notably high (7%). The full mutation frequency of FXS was 7.2% in male patients and 3.3% in female patients. These findings suggest that molecular genetic testing for FXS should be considered in individuals with ID, DD, or ASD of unknown etiology. Increased awareness and determination of the frequency of FXS and its carriers in the general population are essential for examining the feasibility and economic benefits of implementing a population-wide screening program or an appropriate FXS testing program for other specific groups. Enhancing the laboratory infrastructure for prenatal FXS diagnosis in Thailand is crucial for improving accessibility and public health outcomes. Furthermore, expanding genetic counseling services for families affected by FXS is essential to offer better support and guidance, especially in understanding the risks associated with FXS transmission. These initiatives aim to address the diagnostic gap and ensure timely support for individuals with FXS or disorders associated with premutations of the *FMR1* gene.

## Figures and Tables

**Figure 1 ijms-26-07418-f001:**
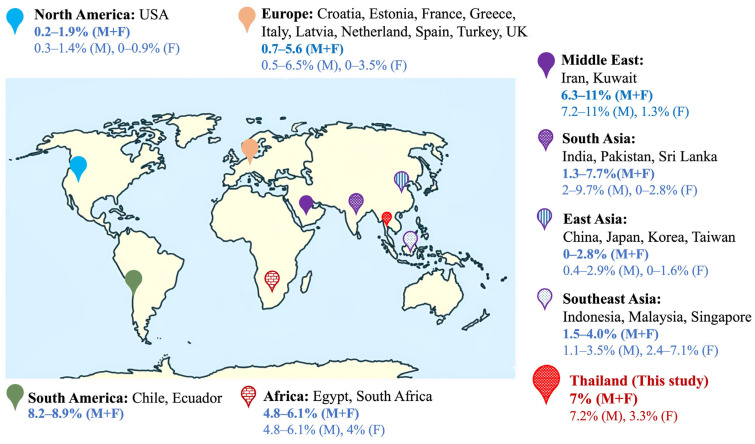
Frequencies of FXS in different high-risk populations. The frequency of fragile X full mutation varies across populations. The frequencies of FXS in different cohorts of Thai patients during various periods were approximately 7%. F, female; M, male.

**Figure 2 ijms-26-07418-f002:**
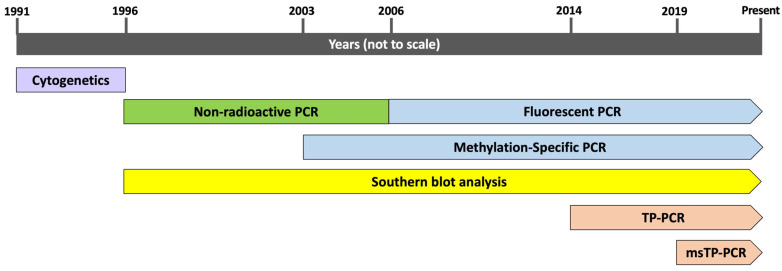
Summary of the techniques used for FXS diagnosis in Thailand during this study period, from 1991 to the present. FXS testing in this study was performed using cytogenetic analysis (1991–1996) and molecular techniques (1996–present). TP-PCR, triplet-repeat-primed PCR; msTP-PCR, methylation-specific triplet-primed PCR.

**Table 1 ijms-26-07418-t001:** Frequencies of FXS in different cohorts studied in Thailand.

Cohort Characteristic	Setting/Region of Thailand	Period of Sample Collection	Source	Total Cases ^a^	No. of Cases with FM (%)	No. of Cases with IM (%)	No. of Cases with NL (%)
Males with ID	Pediatric clinic/Southern Thailand	1991–1999	Limprasert et al., 1999 [24] and unpublished data	Male: 132	Male: 9 (6.8%)	Male: 0	Male: 123 (93.2%)
Males with ID	Pediatric clinic/Central Thailand	1997–1999	Limprasert et al., 1999 [24] and unpublished data	Male: 161	Male: 12 (7.5%)	Male: 0	Male: 149 (92.5%)
Males with ID	Child psychiatric clinic/Northern Thailand	2007–2008	Present study	Male: 101	Male: 8 (7.9%)	Male: 0	Male: 93 (92.1%)
Children in routine FXS testing	Songklanagarind Hospital/All regions of Thailand	2000–2021	Present study	Male: 996 Female: 90	Male: 71 (7.1%) Female: 3 (3.3%)	Male: 1 (0.1%) Female: 0 (0%)	Male: 921 (92.5%) ^b^ Female: 87 (96.7%)
**Total**				Male: 1390 Female: 90 Male + Female: 1480	Male: 100 (7.2%) Female: 3 (3.3%) Male + Female: 103 (7.0%)	Male: 1 (0.07%) Female: 0 (0%) Male + Female: 1 (0.07%)	Male: 1286 (92.5%) Female: 87 (96.7%) Male + Female: 1373 (92.8%)

FXS, fragile X syndrome; FM, full mutation (>200 CGG repeats); IM, intermediate allele; ID, intellectual disability; NL, normal allele. ^a^ Premutation was not identified in the index case of our study. ^b^ Three cases from this cohort exhibited a normal CGG repeat range in the *FMR1* gene. However, they were not included in the group with normal alleles due to sex chromosome abnormalities (two cases of 47,XXY and one case of 46,XY in a female), confirmed by standard karyotyping.

**Table 2 ijms-26-07418-t002:** Prenatal testing results for FXS (12 families).

Family ID of Prenatal Testing	Maternal Alleles	Paternal Allele	*FMR1* CGG Repeats of the Child (C) Alleles and Fetus (P) Alleles
RM10 family	29,PM (80)	29	C1 (M): FM
			C2 (M): mosaic PM (130) and FM
			P1 (M): 29
F4 family	NA	PM (~75)	P1 (M): 29
			P2 (F): 28, PM (~96–130)
F10-2 family	29,PM (75,100)	30	C1 (M): FM
			P1 (F): 29,30
F10-6 family	29,PM (85)	36	C1 (M): FM
			P1 (M): 29
F12-2 family	37,PM (~140)/FM	29	C1 (M): mosaic PM (~101)/313 bp deletion [3]
			C2 (M): FM
			P1 (M): 37
F12-6 family	29,PM (113,167)	29	P1 (F): 29,FM
			P2: mosaic FM as female pattern (47,XXY)
			P3 (F): 29,FM
F20 family	30,FM	NA	C1 (F): 29,FM
			P1 (M): FM
F31 family	32,PM (108)	30	C1 (M): FM
			P1 (M): FM
F33 family	29,PM (104)	29	C1 (M): FM
			P1 (F): 29,29
			P2 (M-twinA): 29
			P3 (M-twinB): 29
F34 family	30,PM (82)	29	C1 (M): FM
			P1 (M): 30
			F2 (M): FM
F35 family	29,PM (100)	29	C1 (M): FM
			C2 (F): 29, ~200
			P1 (F): 29,29
			P2 (F): 29,29
F37 family	23,PM (86)	32	C1 (M): FM
			P1 (F): 23,32

C, child; F, female; M, male; P, fetus with a prenatal diagnosis; PM, premutation; FM, full mutation (>200 CGG repeats) confirmed by methylation PCR and/or Southern blot analysis; NA, not available.

## Data Availability

The original contributions presented in this study are included in the article/Appendix A. Further inquiries can be directed to the corresponding author.

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
