# Peer review of "A 30-Year Experience in Fragile X Syndrome Molecular Diagnosis from a Laboratory in Thailand"

_ijms, 2025, doi:10.3390/ijms26157418_

Round 1

Reviewer 1 Report

Comments and Suggestions for Authors

Fragile X syndrome (FXS) is the most common form of X-linked intellectual disability (ID). This study aimed to share 30 years of experience in diagnosing FXS and determine its frequency in Thailand. The authors retrospectively reviewed 1,480 unrelated patients (1,390 males and 90 females) with ID, developmental delay, or autism spectrum disorder, or those referred for FXS DNA testing at Songklanagarind Hospital, Thailand, over a 30-year period. The samples were analyzed using cytogenetic methods, PCR-based techniques, and/or Southern blot analysis. Full-mutations were identified in 100 males (7.2%) and three females (3.3%). An intermediate allele was detected in one male, while no premutation was found in the index cases. Two males were suspected to have FMR1 gene deletions. Twelve families underwent prenatal testing during the study. Most families undergoing prenatal FXS diagnosis involved mothers who were premutation carriers and had given birth to children affected by FXS. This study represents the largest series of molecular genetic FXS testing cases reported in Thailand. The frequency of FXS identified in different cohorts of Thai patients across various periods was approximately 7%. The authors conclude that this study enhances public awareness of at-risk populations and highlights the importance of prenatal testing and genetic counseling for vulnerable families.

Comments

This is timely written study aimed to determine the frequency of FXS in high-risk groups in Thailand. There are many strengths of this leading FX-diagnostic center in Thailand, including that they used a larger and representative sample, and after 30 years of experience in diagnosing FXS, both postnatal and prenatally. They retrospectively reviewed FXS in 394 male patients with ID across three pediatric cohorts and used a quite large and across the country representative sample: 132 from Southern Thailand, 161 from Central Thailand, and 101 from Northern Thailand. 

Suggestions

A major one is to improve a organization and clarity of this written material to easier follow the narrative. 
For example, the Discussion’s initial paragraph repeats Introduction, and embedded the study findings. 
Lines 100-102, “The overall frequency of full mutations across all cohorts in this study was estimated to be 7% (103/1,480), comprising 7.2% (100/1,390) of males and 3.3% (3/90) of females.”
Lines 176-178 “This study revealed a 4% prevalence of FXS, with females and males accounting for 7.1% (6/84) and 1.1% (1/92) of the cases, respectively.”

Suggestion: The initial paragraph ought to crisply summarize the study findings for a reader to easier follow the flow… for example, we retrospectively reviewed FXS in 394 male patients with ID across three pediatric cohorts using a quite large and across the country representative sample. We found the overall frequency…..and a study prevalence… 

They should also clarify their terms applies frequency, and prevalence. How do they 'complement each other? Do they represent the same?

Then, the Discussion’s next paragraph, to contrast the study findings with the literature in the field.

“Overall, this study revealed a higher than previously reported frequency in male patients with ID, DD, and/or ASD in other Southeast Asian populations (1.1–3.5%), East Asia (0.4–2.9%) and studies from the USA with mixed populations (0.3–1.4%). While the authors account this higher frequency possibly due to the selection selection bias of patients referred to the leading FX-diagnostic center in Thailand, with the majority samples originating from the Southern and Central regions in Thailand, the frequency of fragile X full mutations in Thai male patients is comparable to that reported in South Asia (2–9.7%), the Middle East (7.2–11%), and Africa (6.0–6.4%). A considerable variation in FXS frequency was noted among male patients within European high-risk populations, ranging from 0.5% to 6.5%.”…

And furthermore…

“These findings for prevalence in males are comparable to those results reported in another study of an Indonesian population with DD, where FXS accounted for 1.9% (4/206) of male patients [36]. In Malaysia, among patients with LD from mixed ethnicities, the full FXS mutation was reported in 2.4% (3/127) of females and 3.5% (73/2,057) of males. Similarly, a frequency rate of 2.4% (6/255) was observed among male patients with LD in Singapore. The variation in the reported frequency of fragile X mutations across different Asian studies could be attributed to either selection criteria based on the clinical characteristics of FXS or the influence of a founder effect. However, FXS frequency data from Southeast Asia is sparse and often limited to small-scale studies, highlighting the need for more comprehensive research in this area.”

Then, also add on subheadings. It can be at a beginning of each Discussion paragraph, in italic.
For example, frequency in males. Next one, frequency in females. Next one reflects on CGG repeats..etc

Author Response

Reviewer 1

  1. Comments:

This is timely written study aimed to determine the frequency of FXS in high-risk groups in Thailand. There are many strengths of this leading FX-diagnostic center in Thailand, including that they used a larger and representative sample, and after 30 years of experience in diagnosing FXS, both postnatal and prenatally. They retrospectively reviewed FXS in 394 male patients with ID across three pediatric cohorts and used a quite large and across the country representative sample: 132 from Southern Thailand, 161 from Central Thailand, and 101 from Northern Thailand.

Suggestions:

A major one is to improve a organization and clarity of this written material to easier follow the narrative.

For example, the Discussion’s initial paragraph repeats Introduction, and embedded the study findings.

Lines 100-102, “The overall frequency of full mutations across all cohorts in this study was estimated to be 7% (103/1,480), comprising 7.2% (100/1,390) of males and 3.3% (3/90) of females.”

Lines 176-178 “This study revealed a 4% prevalence of FXS, with females and males accounting for 7.1% (6/84) and 1.1% (1/92) of the cases, respectively.”

Suggestion: The initial paragraph ought to crisply summarize the study findings for a reader to easier follow the flow… for example, we retrospectively reviewed FXS in 394 male patients with ID across three pediatric cohorts using a quite large and across the country representative sample. We found the overall frequency…..and a study prevalence…

Response:

Thank you for your insightful comments and suggestions to improve the organization and clarity of the Discussion section. In response, we have refined the initial paragraph to provide a clear and concise summary of the study findings, ensuring a smoother narrative flow. We have also eliminated repetitive content from the Introduction section, and streamlined the structure to enhance readability and coherence throughout the section.

We appreciate your feedback and believe these revisions effectively address the concerns raised, improving the overall quality of the manuscript..

Page 6, Lines 172-193

We retrospectively reviewed FXS in 1,480 unrelated Thai patients with ID, DD, or ASD across four cohorts, utilizing a large and representative sample covering all regions of Thailand. Our study revealed an overall full mutation frequency of 7%, including 7.2% in males and 3.3% in females, providing valuable insights into the frequency of FXS in this population. The frequency of FXS varies among different populations. Large cohort studies on the frequency of FXS have predominantly been conducted in European and American populations, whereas studies on Asian populations are mostly on a smaller scale [26]. Asia is known for its genetic diversity, with populations across various regions of the continent showing differences in the frequency of FXS. Based on our literature review, a notably high frequency of FXS among individuals with ID has been reported in populations from the Middle East, with the highest frequency (11%) of full mutation observed in Kuwaiti patients [27], potentially due to the founder effect and consanguinity. In contrast, East Asian populations have reported a low frequency of FXS, ranging from 0% to 2.8% [14,28–35]. Very few reports on the frequency of FXS have been published regarding Southeast Asian populations. Previous research conducted in Indonesia reported a frequency of FXS of 4% among patients with ID [36] and 1.5% among patients with DD [37]. In Malaysia, among patients with LD from mixed ethnicities, the full FXS mutation was reported in 3.5% [23]. The variation in the reported frequency of fragile X mutations across different Asian studies could be attributed to either selection criteria based on the clinical characteristics of FXS or the influence of a founder effect [38]. However, FXS frequency data from Southeast Asia is sparse and often limited to small-scale studies, highlighting the need for more comprehensive research in this area.

  1. They should also clarify their terms applies frequency, and prevalence. How do they 'complement each other? Do they represent the same?

Response:

Thank you for your comment. The terms "frequency" and "prevalence" are related but distinct. Frequency refers to the number or proportion of occurrences of a condition within a specific group or population, typically within the context of a study or subset. In contrast, prevalence refers to the proportion of individuals in an entire population who have a specific condition at a given time or over a specified period, offering a broader epidemiological measure. These terms complement each other as frequency provides a snapshot of a condition within a specific context (e.g., study participants), while prevalence captures the condition's reach within a larger population.

To ensure clarity and consistency in our manuscript, we have revised the terminology to use "frequency" throughout, as it aligns more accurately with the scope of our findings and facilitates comparisons with the frequencies reported in previous studies involving specific populations, minimizing potential confusion with population-level metrics implied by prevalence.

  1. Then, the Discussion’s next paragraph, to contrast the study findings with the literature in the field.

“Overall, this study revealed a higher than previously reported frequency in male patients with ID, DD, and/or ASD in other Southeast Asian populations (1.1–3.5%), East Asia (0.4–2.9%) and studies from the USA with mixed populations (0.3–1.4%). While the authors account this higher frequency possibly due to the selection selection bias of patients referred to the leading FX-diagnostic center in Thailand, with the majority samples originating from the Southern and Central regions in Thailand, the frequency of fragile X full mutations in Thai male patients is comparable to that reported in South Asia (2–9.7%), the Middle East (7.2–11%), and Africa (6.0–6.4%). A considerable variation in FXS frequency was noted among male patients within European high-risk populations, ranging from 0.5% to 6.5%.”…

And furthermore…

“These findings for prevalence in males are comparable to those results reported in another study of an Indonesian population with DD, where FXS accounted for 1.9% (4/206) of male patients [36]. In Malaysia, among patients with LD from mixed ethnicities, the full FXS mutation was reported in 2.4% (3/127) of females and 3.5% (73/2,057) of males. Similarly, a frequency rate of 2.4% (6/255) was observed among male patients with LD in Singapore. The variation in the reported frequency of fragile X mutations across different Asian studies could be attributed to either selection criteria based on the clinical characteristics of FXS or the influence of a founder effect. However, FXS frequency data from Southeast Asia is sparse and often limited to small-scale studies, highlighting the need for more comprehensive research in this area.”

Then, also add on subheadings. It can be at a beginning of each Discussion paragraph, in italic.

For example, frequency in males. Next one, frequency in females. Next one reflects on CGG repeats..etc
Response:

Thank you for your valuable feedback. In the revised Discussion section, we first address the overall frequency of FXS full mutations, comparing it with findings from previous studies. Subheadings have been introduced for each key topic to ensure clarity and structure. These include 3.1 Frequency of FXS in males, 3.2 Frequency of FXS in females, 3.3 Distribution of CGG repeats, 3.4 Prenatal FXS diagnosis, 3.5 Full mutation expansion in maternal transmission and AGG interruptions, 3.6 Molecular diagnosis of FXS. For each subheading section, the discussion has been restructured and revised in accordance with the reviewer's suggestions, with improvements made to the sequencing and writing patterns for better coherence and readability.

Page 6, Lines 194-211, Page 7, Lines 212-233

3.1. Frequency of FXS in males

In this study, frequency of FXS in males was found to be 7.2% (100/1,390). Overall, this study revealed a higher than previously reported frequency in male patients with ID, DD, and/or ASD in other Southeast Asian populations (1.1–3.5%) [23,36,37,39], East Asia (0.4–2.9%) [28–30,32–35,40], and studies from the USA with mixed populations (0.3–1.4%) [20,41–43]. Studies in Southeast Asia include a cohort study in Indonesia found full mutation in 1.1% (1/92) of males with ID [36], comparable to another study reporting FXS in 1.9% (4/206) of males patients with DD [37]. In Malaysia, full FXS mutation was reported in 3.5% (73/2,057) of males with LD [23]. A frequency rate of 2.4% (6/255) was observed among male patients with LD in Singapore [39]. The higher frequency observed in our study may be due to the selection bias of patients referred for routine FXS diagnosis based on the most prominent criteria for FXS, which increases the likelihood of identifying individuals with FXS.

However, these findings for frequency of fragile X full mutations in Thai male patients are comparable to that reported in South Asia (2–9.7%) [19,38,44–46], the Middle East (7.2–11%) [16,27], and Africa (6.0–6.4%) [18,47,48]. A considerable variation in FXS frequency was noted among male patients within European high-risk populations, ranging from 0.5% to 6.5% [15,17,49–60] (Figure 1 and Table S2).

3.2. Frequency of FXS in females

The frequency of the fragile X full mutation in Thai females was 3.3% (3/90), which is consistent with several previous studies [18,23,28–30,45,46,49,57]. In contrast, higher frequency rates (7.1%, 6/84) were observed among Indonesian patients with ID, ASD, and/or a family history of FXS [36]; however, this may be due to sample bias, as only a small population of female patients was included in the study. Among females with ID or DD, previous studies have reported varying frequencies ranging from 0% to 7.1% across different populations.

The reported frequency of FXS in female patients is consistent with males' being more affected than females by FXS. However, fewer female than male patients were included in each study, which may not accurately reflect the actual frequency of FXS in females within the population. Although FXS can be explicitly diagnosed, females often exhibit milder features, making early diagnosis more challenging, particularly in regions with limited access to genetic testing, geneticists, and specialized pediatricians. Despite the availability of FXS testing in Thailand for over 30 years, only a few female patients suspected of having FXS have been referred for diagnostic testing. Comparing the frequency of FXS among populations is difficult because the variation in FXS frequency reported in previous studies on high-risk populations is likely due to differences in the populations studied, sample sizes, and diagnostic methods. For instance, PCR followed by gel electrophoresis may not accurately determine the CGG repeat size and may not distinguish between large premutations and full mutations, demonstrating the potential limitations of certain diagnostic methods.

Reviewer 2 Report

Comments and Suggestions for Authors

The current article investigates the natural history of Fragile X syndrome (FXS) in Thailand based upon 30 years’ experience. While the article addresses some important points, some important aspects are missing.

I have some comments that would strengthen the article:

General comments:

1, An important aspect regarding the de novo nature of FXS is missing and this is not made clear in the manuscript. Furthermore, FXS is an X-linked disorder, which this has been limited to one sentence (line 35) and this should be expanded upon. It should be made clear whether the manuscript is alluding to de novo or inherited FXS or both and give the prevalence of these.

2, Is mutation the appropriate term or should pathogenic/likely pathogenic variant be used?

Abstract:

3, Line 16: The authors should clarify what is meant ‘full mutation’.

4, Line 16-17: If mutations were identified in 7.2% of males and 3.3% of females, did the remainder of the cohort have pathogenic de novo variants?

5, Line 23: Does this frequency also consider de novo variants?

Introduction

6, Line 29: I would temper the finding that FXS is the most common cause of inherited intellectual disability (ID). ID is probably caused my 100s of genes and is likely to genetically heterogenous.

7, Line 35: Could non-random XCI (skewing) be responsible for the heterogeneity of clinical severity in FXS as seen in some other X-linked disorders (such as Rett syndrome)?

8, Line 45: What would be the rationale behind of offering the prenatal diagnosis in those with premutation or intermediate alleles? Would this be dependent on family history of the disorder? This section should be explained better. This has in part been alluded to in lines 123-125, 240 but should be also expanded upon here.

9, Line 70: Might come across pejorative and therefore suggest amending the wording “leading”.

10, Line 74: The authors state that the literature was reviewed (also line 170), however, there is no information on how this literature review was done. The authors should consider amending the methodology section of the manuscript to indicate the type of review i.e., scoping/narrative and add information regarding how the current review was conducted i.e., search terms used, databases searched etc. A figure (like a PRISMA diagram) might also be useful.

11, The methods section should come after the introduction.

Results

12, Please check the numbers as it can be difficult for the reader to understand this. Line 13 states that 1,390 male patients were retrospectively reviewed but in line 78, it states that 394 male patients were retrospectively reviewed.

Discussion

13, Suggest the authors add the key findings of the current manuscript to the beginning of the manuscript and then place it in context with other studies.

14, The discussion could also be streamlined. The section would benefit from some sub-headers to break up the text i.e. frequency evidence, pre-natal testing, limitations in methodology etc.

Conclusion

15, Line 448: How useful would it be to have newborn screening programme for FXS? Would there be a risk of false positives? Are all variants in FXS pathogenic?

16, The merits of a newborn screening programme would also depend upon if there were any available treatments for the disorder. At present there are no approved treatment and treatment focuses on the management of symptoms. Would the cost of newborn screening be justified?

Author Response

Reviewer 2

General comments:

1, An important aspect regarding the de novo nature of FXS is missing and this is not made clear in the manuscript. Furthermore, FXS is an X-linked disorder, which this has been limited to one sentence (line 35) and this should be expanded upon. It should be made clear whether the manuscript is alluding to de novo or inherited FXS or both and give the prevalence of these.

Response:

Thank you for your insightful feedback. Fragile X Syndrome is predominantly an inherited disorder caused by a mutation in the FMR1 gene, typically passed down from parents. While most cases of FXS are inherited, a smaller fraction arises from de novo mutations. In this study, almost all cases represented initial FXS testing in index cases. In some situations, further investigation extended to parents and other family members revealed that the majority of FXS full mutations were inherited as expanded alleles from the mother. Furthermore, within our cohort, we identified one case involving a de novo FMR1 deletion.

To address this issue and enhance clarity, we have added more detailed information on the inheritance patterns of FXS in the revised manuscript.

Page 1, Line 41 and Page 2, Lines 42-43

FXS is mainly caused by a repeat expansion mutation in the FMR1 gene, which is usually inherited from the mother, although rare cases of de novo mutations causing FXS have been reported [3,4].

Page 4, Line 142

In the family F12-2, the mother had three children with different FMR1 genetic profiles. The first child had a mosaic premutation and a de novo 313 bp deletion of the FMR1 gene. The second child was a male with a full mutation, and prenatal testing for the third child revealed a male with a normal CGG repeat allele. This family has been reported in our study [3].

2, Is mutation the appropriate term or should pathogenic/likely pathogenic variant be used?

Response:

The most common cause of FXS is a full mutation, defined as more than 200 CGG repeats in the FMR1 gene. This is typically not classified using variant terminology such as pathogenic or likely pathogenic, except in cases of FMR1 point mutations. However, point mutations causing FXS are very rare, often undetectable through routine diagnostic methods for FXS, and are beyond the scope of our research study.

In our study, as well as in most studies of FXS, the ACMG Standards and Guidelines were used, which define categories as follows: full mutation (>200 CGG repeats), premutation (55–200 CGG repeats), intermediate (45–54 CGG repeats), and normal (<45 CGG repeats). To clarify this point, the defined terms for interpretation have already been included at the end of the Materials and Methods section.

Page 12, Lines 458-460

In this study, according to the guidelines of the ACMG [2], FMR1 alleles were classified as normal (< 44 CGG repeats), gray zone or intermediate (45–54 CGG repeats), premutation (55–200 CGG repeats), or full mutation (> 200 CGG repeats).

Abstract: 

3, Line 16: The authors should clarify what is meant ‘full mutation’.

Response:

We have added the defined term of full mutation in the Abstract section following reviewer’s suggestion

Page 1, Line 16

Full mutations (> 200 CGG repeats) were identified in 100 males (7.2%) and three females (3.3%).

4, Line 16-17: If mutations were identified in 7.2% of males and 3.3% of females, did the remainder of the cohort have pathogenic de novo variants?

5, Line 23: Does this frequency also consider de novo variants?

Response:

Thank you for your comment. Although FXS is mainly caused by a repeat expansion mutation in the FMR1 gene, which is usually maternally inherited, in our study not all index cases with full mutations were further tested for the FMR1 gene status in other family members due to limitations in genetic counseling support and financial constraints unique to each family. This restricted our ability to conclusively confirm whether the full mutations observed in all cases were inherited expanded alleles from parents or de novo variants.

Among the mothers of children diagnosed with full FXS mutations who had DNA available for testing, 30 were analyzed for FMR1 gene status. Notably, 28 of these 30 mothers were confirmed to be premutation carriers. This finding supports maternal transmission of full mutation expansion, as discussed in the Results section (Page 5, Lines 151–163) and Table S1. Cases involving de novo variants cannot be completely ruled out; however, based on the tested cohort and available data, maternal inheritance appears to be a predominant mechanism. Moreover, we have addressed the limitation regarding the testing of inheritance within families in the Discussion section (Page 9, Lines 302-304).

We hope this addresses your concern and clarifies our findings within the context of the study design and limitations.

Page 5, Lines 151–163

2.3. Full mutation expansion in maternal transmission

In patients with fragile X full mutation, mothers must typically undergo FMR1 status testing, which can be helpful for management and family planning. Among the mothers of children with full FXS mutations, 30 individuals with available DNA were tested for FMR1 gene status. In this cohort, 28 of 30 mothers of children diagnosed with full mutation were confirmed to be premutation carriers. However, one mother was confirmed to be full mutation carrier with random X-inactivation (29/>200 CGG repeats). Her son was full mutation male, and her daughter was full mutation female with suspected skewed X-inactivation, as only the methylated allele was detected using Southern blot analysis. Another mother was identified as having mosaic premutation and full mutation (37/∼140/>200 CGG repeats), which has been previously reported [3]. None of the mothers carried alleles containing < 70 CGG repeats. TP-PCR analysis revealed 25 full mutation transmissions from maternal alleles with no AGG interruptions and nine transmissions from premutation alleles with one AGG interruption. None of the mothers with premutations had more than one AGG interruption. In this study, the smallest maternal allele expanding to full mutation was 70 CGG repeats with no AGG interruptions. In this cohort, maternal alleles with either no AGG interruptions or one AGG interruption were associated with the expansion of CGG repeats from the premutation to full mutation range in their offspring (Table S1).

Page 9, Lines 302-304

To assess the impact on repeat instability and full mutation expansion, the status of the FMR1 gene and number of AGG interruptions in the mothers of patients with full mutations were determined. For the limitations of our study, not all mothers of cases with full mutations were tested for their FMR1 status due to limited access to genetic counseling and financial constraints faced by each family.

Introduction

6, Line 29: I would temper the finding that FXS is the most common cause of inherited intellectual disability (ID). ID is probably caused my 100s of genes and is likely to genetically heterogenous.

Response:

We agree with your point, and we have revised this section to make it more specific and precise.

Page 1, Lines 29-31

Fragile X syndrome (FXS; MIM 300624) is the most common form of X-linked intellectual disability (ID) and the most common monogenic cause of autism spectrum disorder (ASD).

7, Line 35: Could non-random XCI (skewing) be responsible for the heterogeneity of clinical severity in FXS as seen in some other X-linked disorders (such as Rett syndrome)?

Response:

We agree that non-random X-chromosome inactivation could contribute to the heterogeneity of clinical severity in females with FXS. To address this, we have revised the original sentence for clarity as follows:

Page 1, Lines 35-37

Non-random X-chromosome inactivation (XCI) may explain the milder phenotype observed in females with FXS, with phenotypic variability depending on the degree of non-random XCI [2].

8, Line 45: What would be the rationale behind of offering the prenatal diagnosis in those with premutation or intermediate alleles? Would this be dependent on family history of the disorder? This section should be explained better. This has in part been alluded to in lines 123-125, 240 but should be also expanded upon here.

Response:

We have revised and expanded this section to provide additional details regarding the rationale for offering prenatal diagnosis in individuals with intermediate or premutation alleles, as recommended by the ACMG guidelines. Specifically, the recommendation considers the potential risk of allele instability during transmission, which may result in the expansion to a full mutation associated with fragile X syndrome. This risk is particularly significant for individuals with premutation alleles, though intermediate alleles may also warrant attention, especially in cases with a family history of fragile X-associated disorders.

Page 2, Lines 49-52

According to the American College of Medical Genetics and Genomics (ACMG) Standards and Guidelines for fragile X testing, prenatal diagnosis should be considered in future pregnancies for individuals carrying intermediate or premutation alleles [2,5].

9, Line 70: Might come across pejorative and therefore suggest amending the wording “leading”.

Response:

Thank you for your insightful comment. We have revised the statement to align with your suggestion. The updated phrasing is now: "As the diagnostic center for FXS in Thailand," to accurately reflect our role and maintain consistency in the description.

Page 2, Lines 77-78

As the diagnostic center for FXS in Thailand, receiving samples from all regions of the country, this study shares 30 years of experience in diagnosing FXS, with a particular emphasis on both postnatal and prenatal FXS diagnosis, and aimed to determine the frequency of FXS in high-risk groups in Thailand.

10, Line 74: The authors state that the literature was reviewed (also line 170), however, there is no information on how this literature review was done. The authors should consider amending the methodology section of the manuscript to indicate the type of review i.e., scoping/narrative and add information regarding how the current review was conducted i.e., search terms used, databases searched etc. A figure (like a PRISMA diagram) might also be useful.

Response:

Thank you for your insightful comment regarding the methodology of the literature review. We have add more details in the Methods section of the revised manuscript to clarify that this study is a comprehensive narrative review. The aim of this review was to summarize the frequency of fragile X syndrome across different populations, with a particular focus on high-risk groups and studies utilizing molecular testing.

We searched for relevant articles using two main medical databases: PubMed and Web of Science with the primary keyword “fragile X syndrome,” as associated with the other keywords: FMR1, prevalence, frequency, intellectual disability, developmental delay, autism spectrum disorder, high-risk, and molecular testing. Only articles that were accessible and published in English were selected for the review.

Although the review does not follow a formal systematic review framework, it was designed to be as comprehensive and transparent as possible. This clarification has been incorporated into the revised manuscript. We sincerely appreciate your suggestion and believe that this revision enhances the clarity and rigor of our methodology.

Page 12, Lines 461-469

4.3. Data collection and search strategy for literature review

We conducted a comprehensive narrative review to investigate the frequency of FXS in high-risk groups worldwide using molecular testing. Relevant articles were searched using two main medical databases, PubMed and Web of Science, with the primary keyword “fragile X syndrome,” as combined with the other keywords: “FMR1,” “prevalence,” “frequency,” “intellectual disability,” “developmental delay,” “autism spectrum disorder,” “high-risk,” and “molecular testing."  Only articles that were accessible and published in English were selected for the review. No time limit was set for the published literature. Additional studies were identified through manual screening of reference lists.

11, The methods section should come after the introduction.

Response:

Thank you for your suggestion. However, the structure of this manuscript follows the format and template provided by the journal.

Results

12, Please check the numbers as it can be difficult for the reader to understand this. Line 13 states that 1,390 male patients were retrospectively reviewed but in line 78, it states that 394 male patients were retrospectively reviewed.

Response:

We agree that the statement may cause confusion for the reader, as there appears to be an inconsistency in the numbers reported. To clarify, we have revised the sentence in the Result section.

Page 2, Lines 85-87

We retrospectively reviewed FXS in 1,480 unrelated Thai patients, comprising 394 male patients with ID across three pediatric cohorts, and 1,086 patients referred for routine FXS testing. Among male patients with ID across pediatric cohorts: 132 from Southern Thailand, 161 from Central Thailand, and 101 from Northern Thailand.

Discussion

13, Suggest the authors add the key findings of the current manuscript to the beginning of the manuscript and then place it in context with other studies.

14, The discussion could also be streamlined. The section would benefit from some sub-headers to break up the text i.e. frequency evidence, pre-natal testing, limitations in methodology etc.

Response:

Thank you for your valuable feedback. In the revised Discussion section, we first address the overall frequency of FXS full mutations, comparing it with findings from previous studies (Page 6, Lines 172-193). Subheadings have been introduced for each key topic to ensure clarity and structure. These include 3.1 Frequency of FXS in males, 3.2 Frequency of FXS in females, 3.3 Distribution of CGG repeats, 3.4 Prenatal FXS diagnosis, 3.5 Full mutation expansion in maternal transmission and AGG interruptions, 3.6 Molecular diagnosis of FXS. For each subheading section, the discussion has been restructured and revised in accordance with the reviewer's suggestions, with improvements made to the sequencing and writing patterns for better coherence and readability.

Page 6, Lines 172-193

We retrospectively reviewed FXS in 1,480 unrelated Thai patients with ID, DD, or ASD across four cohorts, utilizing a large and representative sample covering all regions of Thailand. Our study revealed an overall full mutation frequency of 7%, including 7.2% in males and 3.3% in females, providing valuable insights into the frequency of FXS in this population. The frequency of FXS varies among different populations. Large cohort studies on the frequency of FXS have predominantly been conducted in European and American populations, whereas studies on Asian populations are mostly on a smaller scale [26]. Asia is known for its genetic diversity, with populations across various regions of the continent showing differences in the frequency of FXS. Based on our literature review, a notably high frequency of FXS among individuals with ID has been reported in populations from the Middle East, with the highest frequency (11%) of full mutation observed in Kuwaiti patients [27], potentially due to the founder effect and consanguinity. In contrast, East Asian populations have reported a low frequency of FXS, ranging from 0% to 2.8% [14,28–35]. Very few reports on the frequency of FXS have been published regarding Southeast Asian populations. Previous research conducted in Indonesia reported a frequency of FXS of 4% among patients with ID [36] and 1.5% among patients with DD [37]. In Malaysia, among patients with LD from mixed ethnicities, the full FXS mutation was reported in 3.5% [23]. The variation in the reported frequency of fragile X mutations across different Asian studies could be attributed to either selection criteria based on the clinical characteristics of FXS or the influence of a founder effect [38]. However, FXS frequency data from Southeast Asia is sparse and often limited to small-scale studies, highlighting the need for more comprehensive research in this area.

Conclusion

15, Line 448: How useful would it be to have newborn screening programme for FXS? Would there be a risk of false positives? Are all variants in FXS pathogenic?

16, The merits of a newborn screening programme would also depend upon if there were any available treatments for the disorder. At present there are no approved treatment and treatment focuses on the management of symptoms. Would the cost of newborn screening be justified?

Response:

We appreciate the reviewer's feedback and agree with the comment. Thus, we have removed the statement about newborn screening from the Conclusion section and revised the sentence for greater clarity (Page 12, Lines 465-469).

In fact, newborn screening for FXS are somewhat controversial and not widely recommended at this time. Although FXS is a relatively frequent disorder, and there would be the possibility to apply a quite sensitive and specific test, a real benefit of testing is lacking because there is no specific therapy currently to treat children, and treatment is essentially symptom based. To date, newborn screening for FXS has not been implemented in any country, nor has it been supported by governments in newborn screening programs across many countries.

Page 13, Lines 475-479

Increased awareness and determination of the frequency of FXS and its carriers in the general population are essential for examining the feasibility and economic benefits of implementing a population-wide screening program or an appropriate FXS testing program for other specific groups.

Round 2

Reviewer 1 Report

Comments and Suggestions for Authors

The authors have successfully/acceptably addressed all reviewer's suggestions.

Reviewer 2 Report

Comments and Suggestions for Authors

Thank you for considering my suggestions and for revising the manuscript.